# PRUNING CONVOLUTIONAL NEURAL NETWORKS FOR RESOURCE EFFICIENT INFERENCE

**Pavlo Molchanov, Stephen Tyree, Tero Karras, Timo Aila, Jan Kautz**
NVIDIA
`{pmolchanov, styree, tkarras, taila, jkautz}@nvidia.com`

## ABSTRACT

We propose a new formulation for pruning convolutional kernels in neural networks to enable efficient inference. We interleave greedy criteria-based pruning with fine-tuning by backpropagation—a computationally efficient procedure that maintains good generalization in the pruned network. We propose a new criterion based on Taylor expansion that approximates the change in the cost function induced by pruning network parameters. We focus on transfer learning, where large pretrained networks are adapted to specialized tasks. The proposed criterion demonstrates superior performance compared to other criteria, e.g. the norm of kernel weights or feature map activation, for pruning large CNNs after adaptation to fine-grained classification tasks (Birds-200 and Flowers-102) relaying only on the first order gradient information. We also show that pruning can lead to more than $10\times$ theoretical reduction in adapted 3D-convolutional filters with a small drop in accuracy in a recurrent gesture classifier. Finally, we show results for the large-scale ImageNet dataset to emphasize the flexibility of our approach.

## 1 INTRODUCTION

Convolutional neural networks (CNN) are used extensively in computer vision applications, including object classification and localization, pedestrian and car detection, and video classification. Many problems like these focus on specialized domains for which there are only small amounts of carefully curated training data. In these cases, accuracy may be improved by fine-tuning an existing deep network previously trained on a much larger labeled vision dataset, such as images from ImageNet (Russakovsky et al., 2015) or videos from Sports-1M (Karpathy et al., 2014). While transfer learning of this form supports state of the art accuracy, inference is expensive due to the time, power, and memory demanded by the heavyweight architecture of the fine-tuned network.

While modern deep CNNs are composed of a variety of layer types, runtime during prediction is dominated by the evaluation of convolutional layers. With the goal of speeding up inference, we prune entire feature maps so the resulting networks may be run efficiently even on embedded devices. We interleave greedy criteria-based pruning with fine-tuning by backpropagation, a computationally efficient procedure that maintains good generalization in the pruned network.

Neural network pruning was pioneered in the early development of neural networks (Reed, 1993). Optimal Brain Damage (LeCun et al., 1990) and Optimal Brain Surgeon (Hassibi & Stork, 1993) leverage a second-order Taylor expansion to select parameters for deletion, using pruning as regularization to improve training and generalization. This method requires computation of the Hessian matrix partially or completely, which adds memory and computation costs to standard fine-tuning.

In line with our work, Anwar et al. (2015) describe structured pruning in convolutional layers at the level of feature maps and kernels, as well as strided sparsity to prune with regularity within kernels. Pruning is accomplished by particle filtering wherein configurations are weighted by misclassification rate. The method demonstrates good results on small CNNs, but larger CNNs are not addressed.

Han et al. (2015) introduce a simpler approach by fine-tuning with a strong $\ell_2$ regularization term and dropping parameters with values below a predefined threshold. Such unstructured pruning is very effective for network compression, and this approach demonstrates good performance for intra-kernel pruning. But compression may not translate directly to faster inference since modern hardware

exploits regularities in computation for high throughput. So specialized hardware may be needed for efficient inference of a network with intra-kernel sparsity (Han et al., 2016). This approach also requires long fine-tuning times that may exceed the original network training by a factor of 3 or larger. Group sparsity based regularization of network parameters was proposed to penalize unimportant parameters (Wen et al., 2016; Zhou et al., 2016; Alvarez & Salzmann, 2016; Lebedev & Lempitsky, 2016). Regularization-based pruning techniques require per layer sensitivity analysis which adds extra computations. In contrast, our approach relies on global rescaling of criteria for all layers and does not require sensitivity estimation. Moreover, our approach is faster as we directly prune unimportant parameters instead of waiting for their values to be made sufficiently small by optimization under regularization.

Other approaches include combining parameters with correlated weights (Srinivas & Babu, 2015), reducing precision (Gupta et al., 2015; Rastegari et al., 2016) or tensor decomposition (Kim et al., 2015). These approaches usually require a separate training procedure or significant fine-tuning, but potentially may be combined with our method for additional speedups.

## 2 METHOD

The proposed method for pruning consists of the following steps: 1) Fine-tune the network until convergence on the target task; 2) Alternate iterations of pruning and further fine-tuning; 3) Stop pruning after reaching the target trade-off between accuracy and pruning objective, e.g. floating point operations (FLOPs) or memory utilization.

The procedure is simple, but its success hinges on employing the right pruning criterion. In this section, we introduce several efficient pruning criteria and related technical considerations.

Consider a set of training examples $\mathcal{D} = \{\mathcal{X} = \{\mathbf{x}_0, \mathbf{x}_1, ..., \mathbf{x}_N\}, \mathcal{Y} = \{y_0, y_1, ..., y_N\}\}$, where $\mathbf{x}$ and $y$ represent an input and a target output, respectively. The network's parameters[1] $\mathcal{W} = \{(\mathbf{w}_1^1, b_1^1), (\mathbf{w}_1^2, b_1^2), ...(\mathbf{w}_L^{C_\ell}, b_L^{C_\ell})\}$ are optimized to minimize a cost value $\mathcal{C}(\mathcal{D}|\mathcal{W})$. The most common choice for a cost function $\mathcal{C}(\cdot)$ is a negative log-likelihood function. A cost function is selected independently of pruning and depends only on the task to be solved by the original network. In the case of transfer learning, we adapt a large network initialized with parameters $\mathcal{W}_0$ pretrained on a related but distinct dataset.

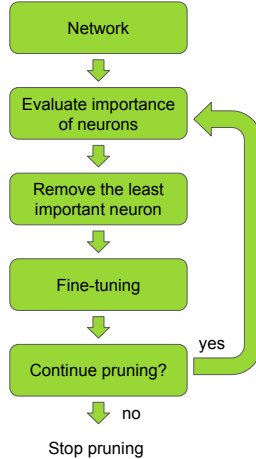

Figure 1: Network pruning as a backward filter.

During pruning, we refine a subset of parameters which preserves the accuracy of the adapted network, $\mathcal{C}(\mathcal{D}|\mathcal{W}') \approx \mathcal{C}(\mathcal{D}|\mathcal{W})$. This corresponds to a combinatorial optimization:

$$\min_{\mathcal{W}'} \left| \mathcal{C}(\mathcal{D}|\mathcal{W}') - \mathcal{C}(\mathcal{D}|\mathcal{W}) \right| \quad \text{s.t.} \quad ||\mathcal{W}'||_0 \leq B, \tag{1}$$

where the $\ell_0$ norm in $||\mathcal{W}'||_0$ bounds the number of non-zero parameters $B$ in $W'$. Intuitively, if $\mathcal{W}' = \mathcal{W}$ we reach the global minimum of the error function, however $||\mathcal{W}'||_0$ will also have its maximum.

Finding a good subset of parameters while maintaining a cost value as close as possible to the original is a combinatorial problem. It will require $2^{|\mathcal{W}|}$ evaluations of the cost function for a selected subset of data. For current networks it would be impossible to compute: for example, VGG-16 has $|\mathcal{W}| = 4224$ convolutional feature maps. While it is impossible to solve this optimization exactly for networks of any reasonable size, in this work we investigate a class of greedy methods. Starting with a full set of parameters $\mathcal{W}$, we iteratively identify and remove the least important parameters, as illustrated in Figure 1. By removing parameters at each iteration, we ensure the eventual satisfaction of the $\ell_0$ bound on $\mathcal{W}'$.

---

[1]A "parameter" $(w, b) \in \mathcal{W}$ might represent an individual weight, a convolutional kernel, or the entire set of kernels that compute a feature map; our experiments operate at the level of feature maps.

Since we focus our analysis on pruning feature maps from convolutional layers, let us denote a set of image feature maps by $\mathbf{z}_\ell \in \mathbb{R}^{H_\ell \times W_\ell \times C_\ell}$ with dimensionality $H_\ell \times W_\ell$ and $C_\ell$ individual maps (or channels).[2] The feature maps can either be the input to the network, $\mathbf{z}_0$, or the output from a convolutional layer, $\mathbf{z}_\ell$ with $\ell \in [1, 2, ..., L]$. Individual feature maps are denoted $\mathbf{z}_\ell^{(k)}$ for $k \in [1, 2, ..., C_\ell]$. A convolutional layer $\ell$ applies the convolution operation $(*)$ to a set of input feature maps $\mathbf{z}_{\ell-1}$ with kernels parameterized by $\mathbf{w}_\ell^{(k)} \in \mathbb{R}^{C_{\ell-1} \times p \times p}$:

$$\mathbf{z}_\ell^{(k)} = \mathbf{g}_\ell^{(k)} \mathcal{R}\big(\mathbf{z}_{\ell-1} * \mathbf{w}_\ell^{(k)} + b_\ell^{(k)}\big), \tag{2}$$

where $\mathbf{z}_\ell^{(k)} \in \mathbb{R}^{H_\ell \times W_\ell}$ is the result of convolving each of $C_{\ell-1}$ kernels of size $p \times p$ with its respective input feature map and adding bias $b_\ell^{(k)}$. We introduce a pruning gate $\mathbf{g}_l \in \{0, 1\}^{C_l}$, an external switch which determines if a particular feature map is included or pruned during feed-forward propagation, such that when $\mathbf{g}$ is vectorized: $\mathcal{W}' = \mathbf{g}\mathcal{W}$.

## 2.1 Oracle pruning

Minimizing the difference in accuracy between the full and pruned models depends on the criterion for identifying the "least important" parameters, called *saliency*, at each step. The best criterion would be an exact empirical evaluation of each parameter, which we denote the *oracle* criterion, accomplished by ablating each non-zero parameter $w \in \mathcal{W}'$ in turn and recording the cost's difference.

We distinguish two ways of using this oracle estimation of importance: 1) *oracle-loss* quantifies importance as the signed change in loss, $\mathcal{C}(\mathcal{D}|\mathcal{W}') - \mathcal{C}(\mathcal{D}|\mathcal{W})$, and 2) *oracle-abs* adopts the absolute difference, $|\mathcal{C}(\mathcal{D}|\mathcal{W}') - \mathcal{C}(\mathcal{D}|\mathcal{W})|$. While both discourage pruning which increases the loss, the oracle-loss version encourages pruning which may decrease the loss, while oracle-abs penalizes any pruning in proportion to its change in loss, regardless of the direction of change.

While the oracle is optimal for this greedy procedure, it is prohibitively costly to compute, requiring $||W'||_0$ evaluations on a training dataset, one evaluation for each remaining non-zero parameter. Since estimation of parameter importance is key to both the accuracy and the efficiency of this pruning approach, we propose and evaluate several criteria in terms of performance and estimation cost.

## 2.2 Criteria for pruning

There are many heuristic criteria which are much more computationally efficient than the oracle. For the specific case of evaluating the importance of a feature map (and implicitly the set of convolutional kernels from which it is computed), reasonable criteria include: the combined $\ell_2$-norm of the kernel weights, the mean, standard deviation or percentage of the feature map's activation, and mutual information between activations and predictions. We describe these criteria in the following paragraphs and propose a new criterion which is based on the Taylor expansion.

**Minimum weight.** Pruning by magnitude of kernel weights is perhaps the simplest possible criterion, and it does not require any additional computation during the fine-tuning process. In case of pruning according to the norm of a set of weights, the criterion is evaluated as: $\Theta_{MW} : \mathbb{R}^{C_{\ell-1} \times p \times p} \to \mathbb{R}$, with $\Theta_{MW}(\mathbf{w}) = \frac{1}{|\mathbf{w}|} \sum_i w_i^2$, where $|\mathbf{w}|$ is dimensionality of the set of weights after vectorization. The motivation to apply this type of pruning is that a convolutional kernel with low $\ell_2$ norm detects less important features than those with a high norm. This can be aided during training by applying $\ell_1$ or $\ell_2$ regularization, which will push unimportant kernels to have smaller values.

**Activation.** One of the reasons for the popularity of the ReLU activation is the sparsity in activation that is induced, allowing convolutional layers to act as feature detectors. Therefore it is reasonable to assume that if an activation value (an output feature map) is small then this feature detector is not important for prediction task at hand. We may evaluate this by mean activation, $\Theta_{MA} :$ $\mathbb{R}^{H_l \times W_\ell \times C_\ell} \to \mathbb{R}$, with $\Theta_{MA}(\mathbf{a}) = \frac{1}{|\mathbf{a}|} \sum_i a_i$ for activation $\mathbf{a} = \mathbf{z}_l^{(k)}$, or by the standard deviation of the activation, $\Theta_{MA\_std}(\mathbf{a}) = \sqrt{\frac{1}{|\mathbf{a}|} \sum_i (a_i - \mu_\mathbf{a})^2}$.

---

[2]While our notation is at times specific to 2D convolutions, the methods are applicable to 3D convolutions, as well as fully connected layers.

**Mutual information.** Mutual information (MI) is a measure of how much information is present in one variable about another variable. We apply MI as a criterion for pruning, $\Theta_{MI} : \mathbb{R}^{H_l \times W_\ell \times C_\ell} \to \mathbb{R}$, with $\Theta_{MI}(\mathbf{a}) = MI(\mathbf{a}, y)$, where $y$ is the target of neural network. MI is defined for continuous variables, so to simplify computation, we exchange it with information gain (IG), which is defined for quantized variables $IG(y|x) = H(x) + H(y) - H(x, y)$, where $H(x)$ is the entropy of variable $x$. We accumulate statistics on activations and ground truth for a number of updates, then quantize the values and compute IG.

**Taylor expansion.** We phrase pruning as an optimization problem, trying to find $\mathcal{W}'$ with bounded number of non-zero elements that minimize $|\Delta C(h_i)| = |\mathcal{C}(\mathcal{D}|\mathcal{W}') - \mathcal{C}(\mathcal{D}|\mathcal{W})|$. With this approach based on the Taylor expansion, we directly approximate change in the loss function from removing a particular parameter. Let $h_i$ be the output produced from parameter $i$. In the case of feature maps, $h = \{z_0^{(1)}, z_0^{(2)}, ..., z_L^{(C_\ell)}\}$. For notational convenience, we consider the cost function equally dependent on parameters and outputs computed from parameters: $\mathcal{C}(\mathcal{D}|h_i) = \mathcal{C}(\mathcal{D}|(\mathbf{w}, b)_i)$. Assuming independence of parameters, we have:

$$\big|\Delta\mathcal{C}(h_i)\big| = \big|\mathcal{C}(\mathcal{D}, h_i = 0) - \mathcal{C}(\mathcal{D}, h_i)\big|, \tag{3}$$

where $\mathcal{C}(\mathcal{D}, h_i = 0)$ is a cost value if output $h_i$ is pruned, while $\mathcal{C}(\mathcal{D}, h_i)$ is the cost if it is not pruned. While parameters are in reality inter-dependent, we already make an independence assumption at each gradient step during training.

To approximate $\Delta\mathcal{C}(h_i)$, we use the first-degree Taylor polynomial. For a function $f(x)$, the Taylor expansion at point $x = a$ is

$$f(x) = \sum_{p=0}^{P} \frac{f^{(p)}(a)}{p!}(x - a)^p + R_p(x), \tag{4}$$

where $f^{(p)}(a)$ is the $p$-th derivative of $f$ evaluated at point $a$, and $R_p(x)$ is the $p$-th order remainder. Approximating $\mathcal{C}(\mathcal{D}, h_i = 0)$ with a first-order Taylor polynomial near $h_i = 0$, we have:

$$\mathcal{C}(\mathcal{D}, h_i = 0) \ = \ \mathcal{C}(\mathcal{D}, h_i) - \frac{\delta\mathcal{C}}{\delta h_i}h_i + R_1(h_i = 0). \tag{5}$$

The remainder $R_1(h_i = 0)$ can be calculated through the Lagrange form:

$$R_1(h_i = 0) = \frac{\delta^2\mathcal{C}}{\delta(h_i^2 = \xi)}\frac{h_i^2}{2}, \tag{6}$$

where $\xi$ is a real number between $0$ and $h_i$. However, we neglect this first-order remainder, largely due to the significant calculation required, but also in part because the widely-used ReLU activation function encourages a smaller second order term. Finally, by substituting Eq. (5) into Eq. (3) and ignoring the remainder, we have $\Theta_{TE} : \mathbb{R}^{H_l \times W_l \times C_l} \to \mathbb{R}^+$, with

$$\Theta_{TE}(h_i) = \big|\Delta\mathcal{C}(h_i)\big| = \left|\mathcal{C}(\mathcal{D}, h_i) - \frac{\delta\mathcal{C}}{\delta h_i}h_i - \mathcal{C}(\mathcal{D}, h_i)\right| = \left|\frac{\delta\mathcal{C}}{\delta h_i}h_i\right|. \tag{7}$$

Intuitively, this criterion prunes parameters that have an almost flat gradient of the cost function w.r.t. feature map $h_i$. This approach requires accumulation of the product of the activation and the gradient of the cost function w.r.t. to the activation, which is easily computed from the same computations for back-propagation. $\Theta_{TE}$ is computed for a multi-variate output, such as a feature map, by

$$\Theta_{TE}(z_l^{(k)}) = \left|\frac{1}{M}\sum_m \frac{\delta C}{\delta z_{l,m}^{(k)}}z_{l,m}^{(k)}\right|, \tag{8}$$

where $M$ is length of vectorized feature map. For a minibatch with $T > 1$ examples, the criterion is computed for each example separately and averaged over $T$.

Independently of our work, Figurnov et al. (2016) came up with similar metric based on the Taylor expansion, called *impact*, to evaluate importance of spatial cells in a convolutional layer. It shows that the same metric can be applied to evaluate importance of different groups of parameters.

**Relation to Optimal Brain Damage.** The Taylor criterion proposed above relies on approximating the change in loss caused by removing a feature map. The core idea is the same as in Optimal Brain Damage (OBD) (LeCun et al., 1990). Here we consider the differences more carefully.

The primary difference is the treatment of the first-order term of the Taylor expansion, in our notation $y = \frac{\delta C}{\delta h} h$ for cost function $C$ and hidden layer activation $h$. After sufficient training epochs, the gradient term tends to zero: $\frac{\delta C}{\delta h} \to 0$ and $\mathbb{E}(y) = 0$. At face value $y$ offers little useful information, hence OBD regards the term as zero and focuses on the second-order term.

However, the *variance* of $y$ is non-zero and correlates with the stability of the local function w.r.t. activation $h$. By considering the absolute change in the cost[3] induced by pruning (as in Eq. 3), we use the absolute value of the first-order term, $|y|$. Under assumption that samples come from independent and identical distribution, $\mathbb{E}(|y|) = \sigma\sqrt{2}/\sqrt{\pi}$ where $\sigma$ is the standard deviation of $y$, known as the expected value of the half-normal distribution. So, while $y$ tends to zero, the expectation of $|y|$ is proportional to the variance of $y$, a value which is empirically more informative as a pruning criterion.

As an additional benefit, we avoid the computation of the second-order Taylor expansion term, or its simplification - diagonal of the Hessian, as required in OBD.

We found important to compare proposed Taylor criteria to OBD. As described in the original papers (LeCun et al., 1990; 1998), OBD can be efficiently implemented similarly to standard back propagation algorithm doubling backward propagation time and memory usage when used together with standard fine-tuning. Efficient implementation of the original OBD algorithm might require significant changes to the framework based on automatic differentiation like Theano to efficiently compute only diagonal of the Hessian instead of the full matrix. Several researchers tried to tackle this problem with approximation techniques (Martens, 2010; Martens et al., 2012). In our implementation, we use efficient way of computing Hessian-vector product (Pearlmutter, 1994) and matrix diagonal approximation proposed by (Bekas et al., 2007), please refer to more details in appendix. With current implementation, OBD is 30 times slower than Taylor technique for saliency estimation, and 3 times slower for iterative pruning, however with different implementation can only be 50% slower as mentioned in the original paper.

**Average Percentage of Zeros (APoZ).** Hu et al. (2016) proposed to explore sparsity in activations for network pruning. ReLU activation function imposes sparsity during inference, and average percentage of positive activations at the output can determine importance of the neuron. Intuitively, it is a good criteria, however feature maps at the first layers have similar APoZ regardless of the network's target as they learn to be Gabor like filters. We will use APoZ to estimate saliency of feature maps.

## 2.3 NORMALIZATION

Some criteria return "raw" values, whose scale varies with the depth of the parameter's layer in the network. A simple layer-wise $\ell_2$-normalization can achieve adequate rescaling across layers:

$$\hat{\Theta}(\mathbf{z}_l^{(k)}) = \frac{\Theta(\mathbf{z}_l^{(k)})}{\sqrt{\sum_j \left(\Theta(\mathbf{z}_l^{(j)})\right)^2}}.$$

## 2.4 FLOPs REGULARIZED PRUNING

One of the main reasons to apply pruning is to reduce number of operations in the network. Feature maps from different layers require different amounts of computation due the number and sizes of input feature maps and convolution kernels. To take this into account we introduce FLOPs regularization:

$$\Theta(\mathbf{z}_l^{(k)}) = \Theta(\mathbf{z}_l^{(k)}) - \lambda \Theta_l^{flops}, \tag{9}$$

where $\lambda$ controls the amount of regularization. For our experiments, we use $\lambda = 10^{-3}$. $\Theta^{flops}$ is computed under the assumption that convolution is implemented as a sliding window (see Appendix). Other regularization conditions may be applied, e.g. storage size, kernel sizes, or memory footprint.

---

[3]OBD approximates the signed difference in loss, while our method approximates absolute difference in loss. We find in our results that pruning based on absolute difference yields better accuracy.

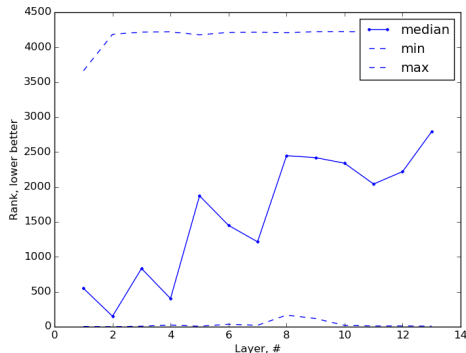

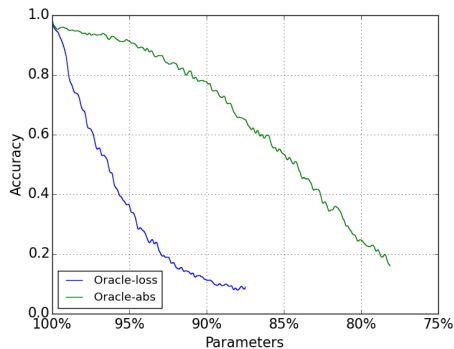

Figure 2: Global statistics of oracle ranking, shown by layer for Birds-200 transfer learning.

Figure 3: Pruning without fine-tuning using oracle ranking for Birds-200 transfer learning.

## 3 RESULTS

We empirically study the pruning criteria and procedure detailed in the previous section for a variety of problems. We focus many experiments on transfer learning problems, a setting where pruning seems to excel. We also present results for pruning large networks on their original tasks for more direct comparison with the existing pruning literature. Experiments are performed within Theano (Theano Development Team, 2016). Training and pruning are performed on the respective training sets for each problem, while results are reported on appropriate holdout sets, unless otherwise indicated. For all experiments we prune *a single feature map* at every pruning iteration, allowing fine-tuning and re-evaluation of the criterion to account for dependency between parameters.

### 3.1 CHARACTERIZING THE ORACLE RANKING

We begin by explicitly computing the oracle for a single pruning iteration of a visual transfer learning problem. We fine-tune the VGG-16 network (Simonyan & Zisserman, 2014) for classification of bird species using the Caltech-UCSD Birds 200-2011 dataset (Wah et al., 2011). The dataset consists of nearly 6000 training images and 5700 test images, covering 200 species. We fine-tune VGG-16 for 60 epochs with learning rate 0.0001 to achieve a test accuracy of 72.2% using uncropped images.

To compute the oracle, we evaluate the change in loss caused by removing each individual feature map from the fine-tuned VGG-16 network. (See Appendix A.3 for additional analysis.) We rank feature maps by their contributions to the loss, where rank 1 indicates the most important feature map—removing it results in the highest increase in loss—and rank 4224 indicates the least important. Statistics of global ranks are shown in Fig. 2 grouped by convolutional layer. We observe: (1) Median global importance tends to decrease with depth. (2) Layers with max-pooling tend to be more important than those without. (VGG-16 has pooling after layers 2, 4, 7, 10, and 13.) However, (3) maximum and minimum ranks show that every layer has some feature maps that are globally important and others that are globally less important. Taken together with the results of subsequent experiments, we opt for encouraging a balanced pruning that distributes selection across all layers.

Next, we iteratively prune the network using pre-computed oracle ranking. In this experiment, we do not update the parameters of the network or the oracle ranking between iterations. Training accuracy is illustrated in Fig. 3 over many pruning iterations. Surprisingly, pruning by smallest absolute change in loss (Oracle-abs) yields higher accuracy than pruning by the net effect on loss (Oracle-loss). Even though the oracle indicates that removing some feature maps individually may decrease loss, instability accumulates due the large absolute changes that are induced. These results support pruning by *absolute* difference in cost, as constructed in Eq. 1.

### 3.2 EVALUATING PROPOSED CRITERIA VERSUS THE ORACLE

To evaluate computationally efficient criteria as substitutes for the oracle, we compute Spearman's rank correlation, an estimate of how well two predictors provide monotonically related outputs,

| | **AlexNet / Flowers-102** | | | | | | **VGG-16 / Birds-200** | | | | | |
| | Weight | Activation | | | OBD | Taylor | Weight | Activation | | | OBD | Taylor | Mutual |
| | | Mean | S.d. | APoZ | | | | Mean | S.d. | APoZ | | | Info. |
| Per layer | 0.17 | 0.65 | 0.67 | 0.54 | 0.64 | **0.77** | 0.27 | 0.56 | 0.57 | 0.35 | 0.59 | **0.73** | 0.28 |
| All layers | 0.28 | 0.51 | 0.53 | 0.41 | 0.68 | 0.37 | 0.34 | 0.35 | 0.30 | 0.43 | 0.65 | 0.14 | 0.35 |
| (w/ $\ell_2$-norm) | 0.13 | 0.63 | 0.61 | 0.60 | - | **0.75** | 0.33 | 0.64 | 0.66 | 0.51 | - | **0.73** | 0.47 |
| | **AlexNet / Birds-200** | | | | | | **VGG-16 / Flowers-102** | | | | | | |
| Per layer | 0.36 | 0.57 | 0.65 | 0.42 | 0.54 | **0.81** | 0.19 | 0.51 | 0.47 | 0.36 | 0.21 | **0.6** | |
| All layers | 0.32 | 0.37 | 0.51 | 0.28 | 0.61 | 0.37 | 0.35 | 0.53 | 0.45 | 0.61 | 0.28 | 0.02 | |
| (w/ $\ell_2$-norm) | 0.23 | 0.54 | 0.57 | 0.49 | - | **0.78** | 0.28 | 0.66 | 0.65 | 0.61 | - | **0.7** | |
| | **AlexNet / ImageNet** | | | | | | | | | | | | |
| Per layer | 0.57 | 0.09 | 0.19 | −0.06 | **0.58** | **0.58** | | | | | | | |
| All layers | 0.67 | 0.00 | 0.13 | −0.08 | **0.72** | 0.11 | | | | | | | |
| (w/ $\ell_2$-norm) | 0.44 | 0.10 | 0.19 | 0.19 | - | 0.55 | | | | | | | |

Table 1: Spearman's rank correlation of criteria vs. oracle for convolutional feature maps of VGG-16 and AlexNet fine-tuned on Birds-200 and Flowers-102 datasets, and AlexNet trained on ImageNet.

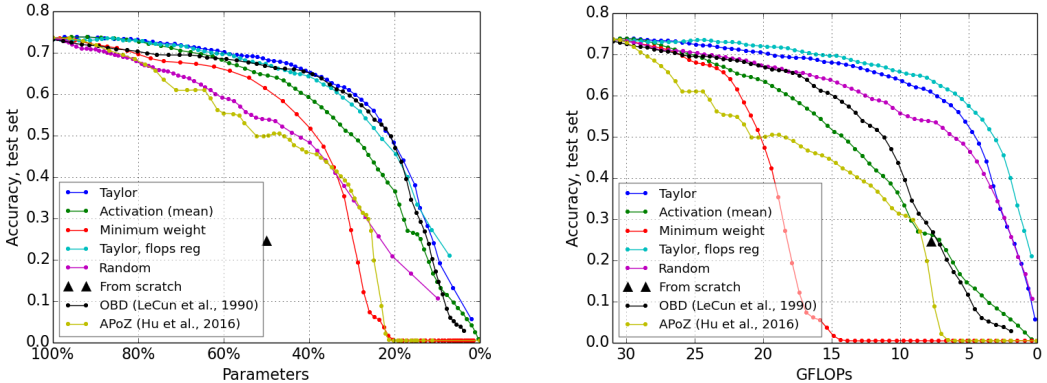

Figure 4: Pruning of feature maps in VGG-16 fine-tuned on the Birds-200 dataset.

even if their relationship is not linear. Given the difference between oracle[4] and criterion ranks $d_i = rank(\Theta_{oracle}(i)) - rank(\Theta_{criterion}(i))$ for each parameter $i$, the rank correlation is computed:

$$\mathcal{S} = 1 - \frac{6}{N(N^2-1)} \sum_{i=1}^{N} d_i^2,  \tag{10}$$

where $N$ is the number of parameters (and the highest rank). This correlation coefficient takes values in $[-1, 1]$, where $-1$ implies full negative correlation, 0 no correlation, and 1 full positive correlation.

We show Spearman's correlation in Table 1 to compare the oracle-abs ranking to rankings by different criteria on a set of networks/datasets some of which are going to be introduced later. Data-dependent criteria (all except weight magnitude) are computed on training data during the fine-tuning before or between pruning iterations. As a sanity check, we evaluate random ranking and observe 0.0 correlation across all layers. "Per layer" analysis shows ranking *within* each convolutional layer, while "All layers" describes ranking across layers. While several criteria do not scale well across layers with raw values, a layer-wise $\ell_2$-normalization significantly improves performance. The Taylor criterion has the highest correlation among the criteria, both within layers and across layers (with $\ell_2$ normalization). OBD shows the best correlation across layers when no normalization used; it also shows best results for correlation on ImageNet dataset. (See Appendix A.2 for further analysis.)

### 3.3 Pruning fine-tuned ImageNet networks

We now evaluate the full iterative pruning procedure on two transfer learning problems. We focus on reducing the number of convolutional feature maps and the total estimated floating point operations (FLOPs). Fine-grained recognition is difficult for relatively small datasets without relying on transfer

---
[4]We use Oracle-abs because of better performance in previous experiment

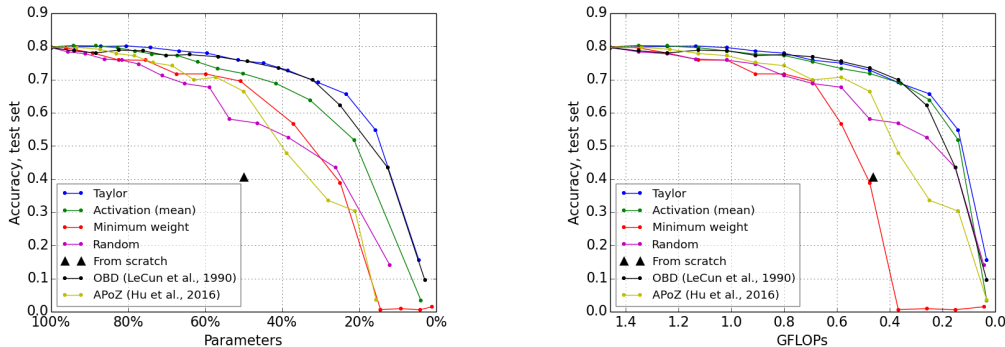

Figure 5: Pruning of feature maps in AlexNet on fine-tuned on Flowers-102.

learning. Branson et al. (2014) show that training CNN from scratch on the Birds-200 dataset achieves test accuracy of only 10.9%. We compare results to training a randomly initialized CNN with half the number of parameters per layer, denoted "from scratch".

Fig. 4 shows pruning of VGG-16 after fine-tuning on the Birds-200 dataset (as described previously). At each pruning iteration, we remove a single feature map and then perform 30 minibatch SGD updates with batch-size 32, momentum 0.9, learning rate $10^{-4}$, and weight decay $10^{-4}$. The figure depicts accuracy relative to the pruning rate (left) and estimated GFLOPs (right). The Taylor criterion shows the highest accuracy for nearly the entire range of pruning ratios, and with FLOPs regularization demonstrates the best performance relative to the number of operations. OBD shows slightly worse performance of pruning in terms of parameters, however significantly worse in terms of FLOPs.

In Fig. 5, we show pruning of the CaffeNet implementation of AlexNet (Krizhevsky et al., 2012) after adapting to the Oxford Flowers 102 dataset (Nilsback & Zisserman, 2008), with 2040 training and 6129 test images from 102 species of flowers. Criteria correlation with oracle-abs is summarized in Table 1. We initially fine-tune the network for 20 epochs using a learning rate of 0.001, achieving a final test accuracy of 80.1%. Then pruning procedes as previously described for Birds-200, except with only 10 mini-batch updates between pruning iterations. We observe the superior performance of the Taylor and OBD criteria in both number of parameters and GFLOPs.

We observed that Taylor criterion shows the best performance which is closely followed by OBD with a bit lower Spearman's rank correlation coefficient. Implementing OBD takes more effort because of computation of diagonal of the Hessian and it is 50% to 300% slower than Taylor criteria that relies on first order gradient only.

Fig. 6 shows pruning with the Taylor technique and a varying number of fine-tuning updates between pruning iterations. Increasing the number of updates results in higher accuracy, but at the cost of additional runtime of the pruning procedure.

During pruning we observe a small drop in accuracy. One of the reasons is fine-tuning between pruning iterations. Accuracy of the initial network can be improved with longer fine tunning and search of better optimization parameters. For example accuracy of unpruned VGG16 network on Birds-200 goes up to 75% after extra 128k updates. And AlexNet on Flowers-102 goes up to 82.9% after 130k updates. It should be noted that with farther fine-tuning of pruned networks we can achieve higher accuracy as well, therefore the one-to-one comparison of accuracies is rough.

### 3.4 PRUNING A RECURRENT 3D-CNN NETWORK FOR HAND GESTURE RECOGNITION

Molchanov et al. (2016) learn to recognize 25 dynamic hand gestures in streaming video with a large recurrent neural network. The network is constructed by adding recurrent connections to a 3D-CNN pretrained on the Sports-1M video dataset (Karpathy et al., 2014) and fine tuning on a gesture dataset. The full network achieves an accuracy of 80.7% when trained on the depth modality, but a single inference requires an estimated 37.8 GFLOPs, too much for deployment on an embedded GPU. After several iterations of pruning with the Taylor criterion with learning rate 0.0003, momentum 0.9, FLOPs regularization $10^{-3}$, we reduce inference to 3.0 GFLOPs, as shown in Fig. 7. While pruning

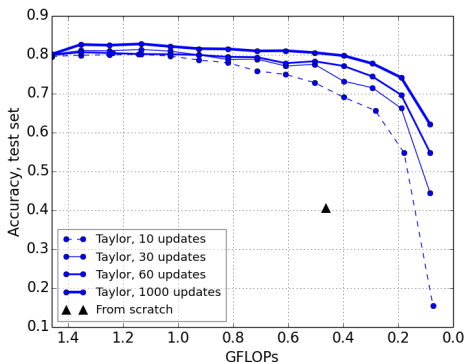

Figure 6: Varying the number of minibatch updates between pruning iterations with AlexNet/Flowers-102 and the Taylor criterion.

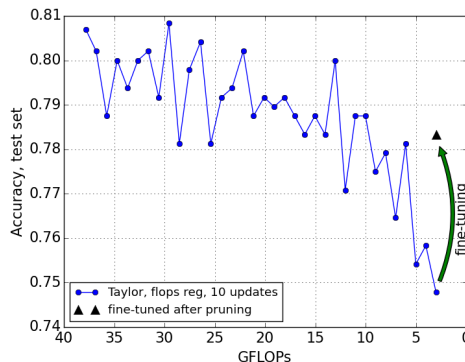

Figure 7: Pruning of a recurrent 3D-CNN for dynamic hand gesture recognition (Molchanov et al., 2016).

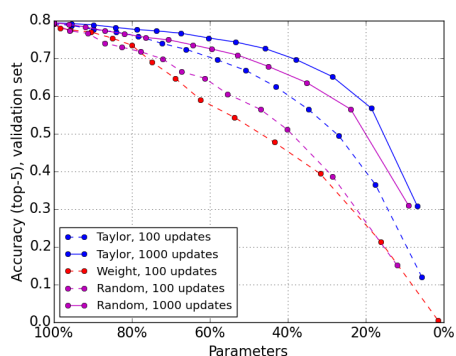

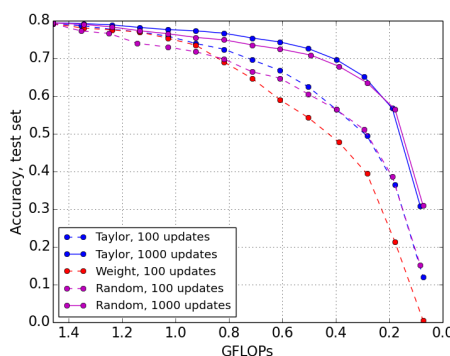

Figure 8: Pruning of AlexNet on Imagenet with varying number of updates between pruning iterations.

increases classification error by nearly $6\%$, additional fine-tuning restores much of the lost accuracy, yielding a final pruned network with a $12.6\times$ reduction in GFLOPs and only a $2.5\%$ loss in accuracy.

## 3.5 PRUNING NETWORKS FOR IMAGENET

We also test our pruning scheme on the large-scale ImageNet classification task. In the first experiment, we begin with a trained CaffeNet implementation of AlexNet with $79.2\%$ top-5 validation accuracy. Between pruning iterations, we fine-tune with learning rate $10^{-4}$, momentum 0.9, weight decay $10^{-4}$, batch size 32, and drop-out $50\%$. Using a subset of 5000 training images, we compute oracle-abs and Spearman's rank correlation with the criteria, as shown in Table 1. Pruning traces are illustrated in Fig. 8. We observe: 1) Taylor performs better than random or minimum weight pruning when 100 updates are used between pruning iterations. When results are displayed w.r.t. FLOPs, the difference with random pruning is only $0\%-4\%$, but the difference is higher, $1\%-10\%$, when plotted with the number of feature maps pruned. 2) Increasing the number of updates from 100 to 1000 improves performance of pruning significantly for both the Taylor criterion and random pruning.

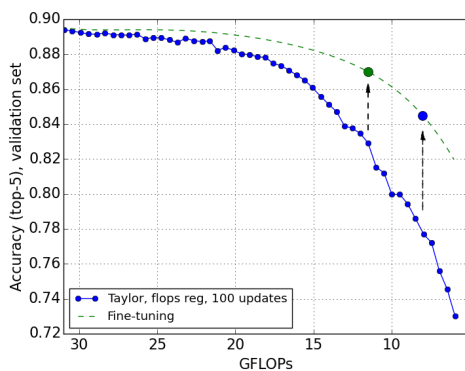

Figure 9: Pruning of the VGG-16 network on ImageNet, with additional following fine-tuning at 11.5 and 8 GFLOPs.

| Hardware | Batch | Accuracy | Time, ms | Accuracy | Time (speed up) | Accuracy | Time (speed up) |
|---|---|---|---|---|---|---|---|
| **AlexNet / Flowers-102**, 1.46 GFLOPs | | | | *41% feature maps, 0.4 GFLOPs* | | *19.5% feature maps, 0.2 GFLOPs* | |
| CPU: Intel Core i7-5930K | 16 | 80.1% | 226.4 | 79.8%(-0.3%) | 121.4 (1.9x) | 74.1%(-6.0%) | 87.0 (2.6x) |
| GPU: GeForce GTX TITAN X (Pascal) | 16 | | 4.8 | | 2.4 (2.0x) | | 1.9 (2.5x) |
| GPU: GeForce GTX TITAN X (Pascal) | 512 | | 88.3 | | 36.6 (2.4x) | | 27.4 (3.2x) |
| GPU: NVIDIA Jetson TX1 | 32 | | 169.2 | | 73.6 (2.3x) | | 58.6 (2.9x) |
| **VGG-16 / ImageNet**, 30.96 GFLOPs | | | | *66% feature maps, 11.5 GFLOPs* | | *52% feature maps, 8.0 GFLOPs* | |
| CPU: Intel Core i7-5930K | 16 | 89.3% | 2564.7 | 87.0% (-2.3%) | 1483.3 (1.7x) | 84.5% (-4.8%) | 1218.4 (2.1x) |
| GPU: GeForce GTX TITAN X (Pascal) | 16 | | 68.3 | | 31.0 (2.2x) | | 20.2 (3.4x) |
| GPU: NVIDIA Jetson TX1 | 4 | | 456.6 | | 182.5 (2.5x) | | 138.2 (3.3x) |
| **R3DCNN / nvGesture**, 37.8 GFLOPs | | | | *25% feature maps, 3 GFLOPs* | | | |
| GPU: GeForce GT 730M | 1 | 80.7% | 438.0 | 78.2% (-2.5%) | 85.0 (5.2x) | | |

Table 2: Actual speed up of networks pruned by Taylor criterion for various hardware setup. All measurements were performed with PyTorch with cuDNN v5.1.0, except R3DCNN which was implemented in C++ with cuDNN v4.0.4). Results for ImageNet dataset are reported as top-5 accuracy on validation set. Results on AlexNet / Flowers-102 are reported for pruning with 1000 updates between iterations and no fine-tuning after pruning.

For a second experiment, we prune a trained VGG-16 network with the same parameters as before, except enabling FLOPs regularization. We stop pruning at two points, 11.5 and 8.0 GFLOPs, and fine-tune both models for an additional five epochs with learning rate $10^{-4}$. Fine-tuning after pruning significantly improves results: the network pruned to 11.5 GFLOPs improves from 83% to 87% top-5 validation accuracy, and the network pruned to 8.0 GFLOPs improves from 77.8% to 84.5%.

### 3.6 SPEED UP MEASUREMENTS

During pruning we were measuring reduction in computations by FLOPs, which is a common practice (Han et al., 2015; Lavin, 2015a;b). Improvements in FLOPs result in monotonically decreasing inference time of the networks because of removing entire feature map from the layer. However, time consumed by inference dependents on particular implementation of convolution operator, parallelization algorithm, hardware, scheduling, memory transfer rate etc. Therefore we measure improvement in the inference time for selected networks to see real speed up compared to unpruned networks in Table 2. We observe significant speed ups by proposed pruning scheme.

## 4 CONCLUSIONS

We propose a new scheme for iteratively pruning deep convolutional neural networks. We find: 1) CNNs may be successfully pruned by iteratively removing the least important parameters—feature maps in this case—according to heuristic selection criteria; 2) a Taylor expansion-based criterion demonstrates significant improvement over other criteria; 3) per-layer normalization of the criterion is important to obtain global scaling.

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

# A APPENDIX

## A.1 FLOPs COMPUTATION

To compute the number of floating-point operations (FLOPs), we assume convolution is implemented as a sliding window and that the nonlinearity function is computed for free. For convolutional kernels we have:

$$\text{FLOPs} = 2HW(C_{in}K^2 + 1)C_{out}, \tag{11}$$

where $H$, $W$ and $C_{in}$ are height, width and number of channels of the input feature map, $K$ is the kernel width (assumed to be symmetric), and $C_{out}$ is the number of output channels.

For fully connected layers we compute FLOPs as:

$$\text{FLOPs} = (2I - 1)O, \tag{12}$$

where $I$ is the input dimensionality and $O$ is the output dimensionality.

We apply FLOPs regularization during pruning to prune neurons with higher FLOPs first. FLOPs per convolutional neuron in every layer:

$$\text{VGG16: } \Theta^{flops} = [3.1, 57.8, 14.1, 28.9, 7.0, 14.5, 14.5, 3.5, 7.2, 7.2, 1.8, 1.8, 1.8, 1.8]$$
$$\text{AlexNet: } \Theta^{flops} = [2.3, 1.7, 0.8, 0.6, 0.6]$$
$$\text{R3DCNN: } \Theta^{flops} = [5.6, 86.9, 21.7, 43.4, 5.4, 10.8, 1.4, 1.4]$$

## A.2 NORMALIZATION ACROSS LAYERS

Scaling a criterion across layers is very important for pruning. If the criterion is not properly scaled, then a hand-tuned multiplier would need to be selected for each layer. Statistics of feature map ranking by different criteria are shown in Fig. 10. Without normalization (Fig. 14a–14d), the weight magnitude criterion tends to rank feature maps from the first layers more important than last layers; the activation criterion ranks middle layers more important; and Taylor ranks first layers higher. After $\ell_2$ normalization (Fig. 10d–10f), all criteria have a shape more similar to the oracle, where each layer has some feature maps which are highly important and others which are unimportant.

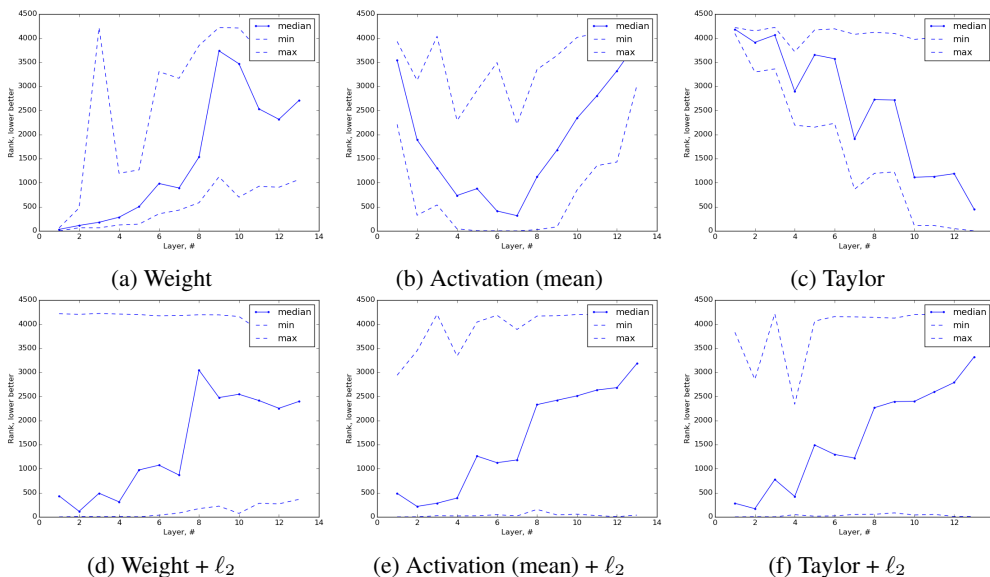

(a) Weight   (b) Activation (mean)   (c) Taylor

(d) Weight + $\ell_2$   (e) Activation (mean) + $\ell_2$   (f) Taylor + $\ell_2$

Figure 10: Statistics of feature map ranking by raw criteria values (top) and by criteria values after $\ell_2$ normalization (bottom).

|  | MI | Weight | Activation | | | OBD | Taylor |
|---|---|---|---|---|---|---|---|
|  |  |  | Mean | S.d. | APoZ |  |  |
| **Per layer** | | | | | | | |
| Layer 1 | 0.41 | 0.40 | 0.65 | 0.78 | 0.36 | 0.54 | **0.95** |
| Layer 2 | 0.23 | 0.57 | 0.56 | 0.59 | 0.33 | 0.78 | **0.90** |
| Layer 3 | 0.14 | 0.55 | 0.48 | 0.45 | 0.51 | 0.66 | **0.74** |
| Layer 4 | 0.26 | 0.23 | 0.58 | 0.42 | 0.10 | 0.36 | **0.80** |
| Layer 5 | 0.17 | 0.28 | 0.49 | 0.52 | 0.15 | 0.54 | **0.69** |
| Layer 6 | 0.21 | 0.18 | 0.41 | 0.48 | 0.16 | 0.49 | **0.63** |
| Layer 7 | 0.12 | 0.19 | 0.54 | 0.49 | 0.38 | 0.55 | **0.71** |
| Layer 8 | 0.18 | 0.23 | 0.43 | 0.42 | 0.30 | 0.50 | **0.54** |
| Layer 9 | 0.21 | 0.18 | 0.50 | 0.55 | 0.35 | 0.53 | **0.61** |
| Layer 10 | 0.26 | 0.15 | 0.59 | 0.60 | 0.45 | 0.61 | **0.66** |
| Layer 11 | 0.41 | 0.12 | 0.61 | 0.65 | 0.45 | 0.64 | **0.72** |
| Layer 12 | 0.47 | 0.15 | 0.60 | 0.66 | 0.39 | 0.66 | **0.72** |
| Layer 13 | 0.61 | 0.21 | **0.77** | 0.76 | 0.65 | 0.76 | **0.77** |
| Mean | 0.28 | 0.27 | 0.56 | 0.57 | 0.35 | 0.59 | **0.73** |
| **All layers** | | | | | | | |
| No normalization | 0.35 | 0.34 | 0.35 | 0.30 | 0.43 | 0.65 | 0.14 |
| $\ell_1$ normalization | 0.47 | 0.37 | 0.63 | 0.63 | 0.52 | 0.65 | 0.71 |
| $\ell_2$ normalization | 0.47 | 0.33 | 0.64 | 0.66 | 0.51 | 0.60 | **0.73** |
| Min-max normalization | 0.27 | 0.17 | 0.52 | 0.57 | 0.42 | 0.54 | 0.67 |

Table 3: Spearman's rank correlation of criteria vs oracle-abs in VGG-16 fine-tuned on Birds 200.

### A.3 ORACLE COMPUTATION FOR VGG-16 ON BIRDS-200

We compute the change in the loss caused by removing individual feature maps from the VGG-16 network, after fine-tuning on the Birds-200 dataset. Results are illustrated in Fig. 11a-11b for each feature map in layers 1 and 13, respectively. To compute the oracle estimate for a feature map, we remove the feature map and compute the network prediction for each image in the training set using the central crop with no data augmentation or dropout. We draw the following conclusions:

- The contribution of feature maps range from positive (above the red line) to slightly negative (below the red line), implying the existence of some feature maps which decrease the training cost when removed.

- There are many feature maps with little contribution to the network output, indicated by almost zero change in loss when removed.

- Both layers contain a small number of feature maps which induce a significant increase in the loss when removed.

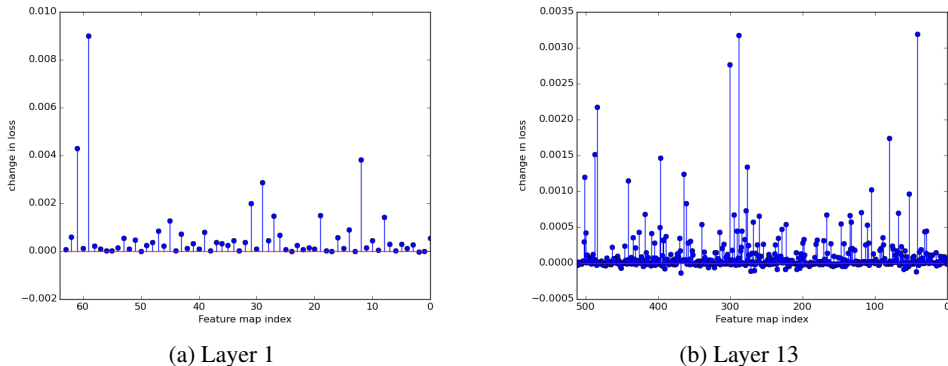

(a) Layer 1                              (b) Layer 13

Figure 11: Change in training loss as a function of the removal of a single feature map from the VGG-16 network after fine-tuning on Birds-200. Results are plotted for two convolutional layers w.r.t. the index of the removed feature map index. The loss with all feature maps, 0.00461, is indicated with a red horizontal line.

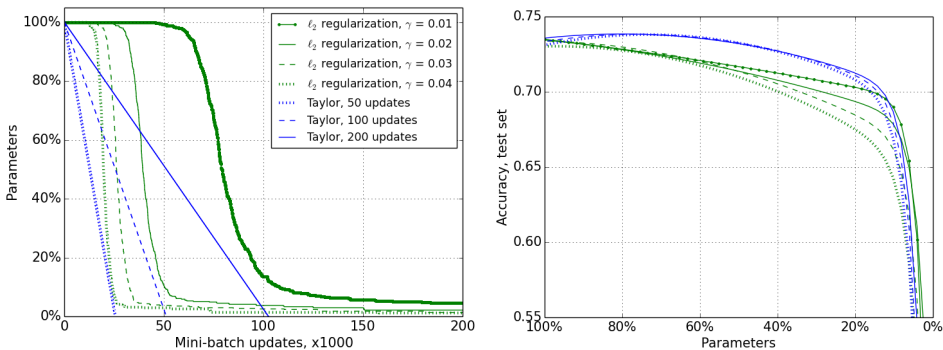

Figure 12: Comparison of our iterative pruning with pruning by regularization

Table 3 contains a layer-by-layer listing of Spearman's rank correlation of several criteria with the ranking of oracle-abs. In this more detailed comparison, we see the Taylor criterion shows higher correlation for all individual layers. For several methods including Taylor, the worst correlations are observed for the middle of the network, layers 5-10. We also evaluate several techniques for normalization of the raw criteria values for comparison across layers. The table shows the best performance is obtained by $\ell_2$ normalization, hence we select it for our method.

### A.4 COMPARISON WITH WEIGHT REGULARIZATION

Han et al. (2015) find that fine-tuning with high $\ell_1$ or $\ell_2$ regularization causes unimportant connections to be suppressed. Connections with energy lower than some threshold can be removed on the assumption that they do not contribute much to subsequent layers. The same work also finds that thresholds must be set separately for each layer depending on its sensitivity to pruning. The procedure to evaluate sensitivity is time-consuming as it requires pruning layers independently during evaluation.

The idea of pruning with high regularization can be extended to removing the kernels for an entire feature map if the $\ell_2$ norm of those kernels is below a predefined threshold. We compare our approach with this regularization-based pruning for the task of pruning the last convolutional layer of VGG-16 fine-tuned for Birds-200. By considering only a single layer, we avoid the need to compute layerwise sensitivity. Parameters for optimization during fine-tuning are the same as other experiments with the Birds-200 dataset. For the regularization technique, the pruning threshold is set to $\sigma = 10^{-5}$ while we vary the regularization coefficient $\gamma$ of the $\ell_2$ norm on each feature map kernel.[5] We prune only kernel weights, while keeping the bias to maintain the same expected output.

A comparison between pruning based on regularization and our greedy scheme is illustrated in Fig. 12. We observe that our approach has higher test accuracy for the same number of remaining unpruned feature maps, when pruning $85\%$ or more of the feature maps. We observe that with high regularization all weights tend to zero, not only unimportant weights as Han et al. (2015) observe in the case of ImageNet networks. The intuition here is that with regularization we push all weights down and potentially can affect important connections for transfer learning, whereas in our iterative procedure we only remove unimportant parameters leaving others untouched.

### A.5 COMBINATION OF CRITERIA

One of the possibilities to improve saliency estimation is to combine several criteria together. One of the straight forward combinations is Taylor and mean activation of the neuron. We compute the joint criteria as $\Theta_{joint}(\mathbf{z}_l^{(k)}) = (1-\lambda)\hat{\Theta}_{Taylor}(\mathbf{z}_l^{(k)}) + \lambda\hat{\Theta}_{Activation}(\mathbf{z}_l^{(k)})$ and perform a grid search of parameter $\lambda$ in Fig.13. The highest correlation value for each dataset is marked with with vertical bar with $\lambda$ and gain. We observe that the gain of linearly combining criteria is negligibly small (see $\Delta$'s in the figure).

---

[5]In our implementation, the regularization coefficient is multiplied by the learning rate equal to $10^{-4}$.

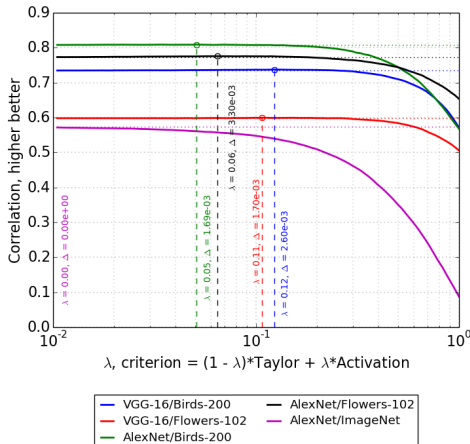

Figure 13: Spearman rank correlation for linear combination of criteria. The *per layer* metric is used. Each $\Delta$ indicates the gain in correlation for one experiment.

## A.6 Optimal Brain Damage implementation

OBD computes saliency of a parameter by computing a product of the squared magnitude of the parameter and the corresponding element on the diagonal of the Hessian. For many deep learning frameworks, an efficient implementation of the diagonal evaluation is not straightforward and approximation techniques must be applied. Our implementation of Hessian diagonal computation was inspired by Dauphin et al. (2015) work, where the technique proposed by Bekas et al. (2007) was used to evaluate SGD preconditioned with the Jacobi preconditioner. It was shown that diagonal of the Hessian can be approximated as:

$$\text{diag}(\mathbf{H}) = \mathbb{E}[\mathbf{v} \odot \mathbf{H}\mathbf{v}] = \mathbb{E}[\mathbf{v} \odot \nabla(\nabla \mathcal{C} \cdot \mathbf{v})], \tag{13}$$

where $\odot$ is the element-wise product, $\mathbf{v}$ are random vectors with entries $\pm 1$, and $\nabla$ is the gradient operator. To compute saliency with OBD, we randomly draw $\mathbf{v}$ and compute the diagonal over 10 iterations for a single minibatch for 1000 mini batches. We found that this number of mini batches is required to compute close approximation of the Hessian's diagonal (which we verified). Computing saliency this way is computationally expensive for iterative pruning, and we use a slightly different but more efficient procedure. Before the first pruning iteration, saliency is initialized from values computed off-line with 1000 minibatches and 10 iterations, as described above. Then, at every minibatch we compute the OBD criteria with only one iteration and apply an exponential moving averaging with a coefficient of 0.99. We verified that this computes a close approximation to the Hessian's diagonal.

## A.7 Correlation of Taylor criterion with gradient and activation

The Taylor criterion is composed of both an activation term and a gradient term. In Figure 14, we depict the correlation between the Taylor criterion and each constituent part. We consider expected absolute value of the gradient instead of the mean, because otherwise it tends to zero. The plots are computed from pruning criteria for an unpruned VGG network fine-tuned for the Birds-200 dataset. (Values are shown after layer-wise normalization). Figure 14(a-b) depict the Taylor criterion in the y-axis for all neurons w.r.t. the gradient and activation components, respectively. The bottom $10\%$ of neurons (lowest Taylor criterion, most likely to be pruned) are depicted in red, while the top $10\%$ are shown in green. Considering all neurons, both gradient and activation components demonstrate a linear trend with the Taylor criterion. However, for the bottom $10\%$ of neurons, as shown in Figure 14(c-d), the activation criterion shows much stronger correlation, with lower activations indicating lower Taylor scores.

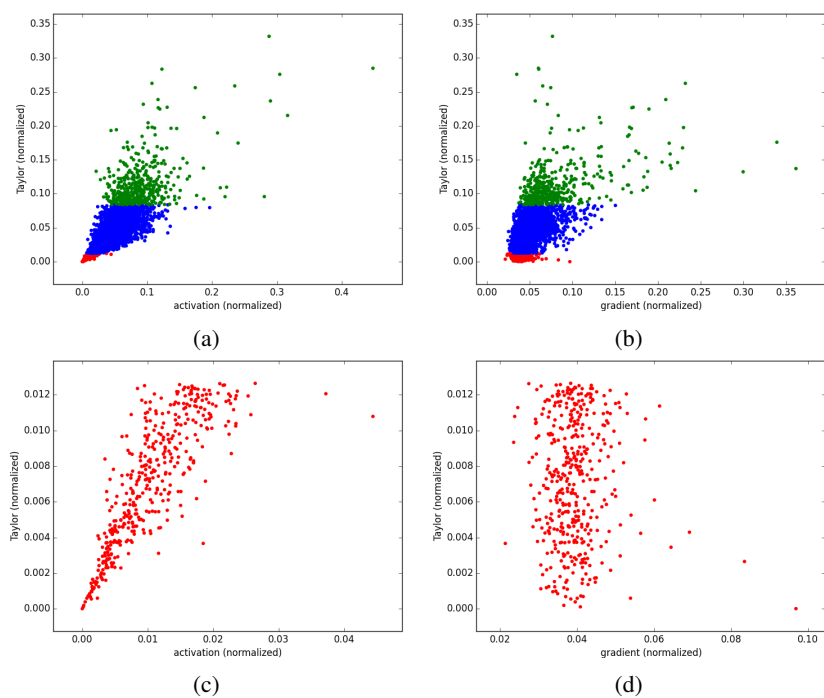

Figure 14: Correlation of Taylor criterion with gradient and activation (after layer-wise $\ell_2$ normalization) for all neurons (a-b) and bottom $10\%$ of neurons (c-d) for unpruned VGG after fine-tuning on Birds-200.

