# Peer review of "Pruning Convolutional Neural Networks for Resource Efficient Inference"

_ICLR 2017 — accepted_

[Public Comment · Zehao Huang · 09 Nov 2016]
**Some questions about the experiments**

This is a very good paper about neurons pruning. The authors focus on the problem about pruning convolutional kernels. 

I have some questions about the experiments in this paper. 

1) How to decide the number of neurons for pruning in each iteration ?

Is there any criterions to decide the number of neurons for purning in each iteration automatically?

2) Did the authors only pruning convolution kernels in the experiments?

In section 3.4 and 3.5, the authors showed the pruning results measured by parameters and GFLOPs. Did they only compute the parameters and GFLOPs of convolution layers or both convolution layers and fully connected layers?

Thanks

[Official Review · AnonReviewer1 · rating 9 · confidence 4 · 16 Dec 2016]
**Strong experimental evaluation of a theoretically justified pruning method**
soundness 5 · originality 3

This paper presents a novel way of pruning filters from convolutional neural networks with a strong theoretical justification. The proposed methods is derived from the first order Taylor expansion of the loss change while pruning a particular unit. This leads to simple weighting of the unit activation with its gradient w.r.t. loss function and performs better than simply using the activation magnitude as the heuristic for pruning. This intuitively makes sense, as we would like to remove not only the filters with low activation, but also filters where the incorrect activation value would not have small influence on the target loss.

Authors thoroughly investigate multiple baselines, including an oracle which sets an upper bound on the target performance even though it is computationally expensive. The devised method seems to be quite elegant and authors show that it generalizes well on multiple tasks and is computationally more than feasible as it is easy to combine with traditional fine tuning procedure. Also, the work clearly shows the trade-offs of increased speed and decreased performance, which is useful for practical applications.

It would be also useful to compare against different baselines, e.g. [1]. However this method seems to be more useful as it does not involve training of a new network (and thus is probably much faster).

Suggestion - maybe it can be extended in the future towards also removing only parts of the filters(e.g. for the 3D convolution)? This may be more complicated as it would need to change the implementation of convolution operator, but can lead to further speedup.

[1]

[Official Review · AnonReviewer4 · rating 6 · confidence 4 · 16 Dec 2016 (modified: 20 Jan 2017)]

Authors propose a neural pruning technique starting from trained models using an approximation of change in the cost function and outperform other criteria. Authors obtain solid speedups while maintaining reasonable accuracy thanks to finetuning after pruning. Comparisons to existing methods is weak as GFLOPS graphs only show a couple simple baselines and no prior work baselines. I would be more convinced of the superiority of the approach with such comparison.

[Official Review · AnonReviewer3 · rating 7 · confidence 4 · 22 Dec 2016 (modified: 27 Jan 2017)]
**Empirically justified pruning strategy, a few missing comparisons**
soundness 3 · impact 4

Authors propose a strategy for pruning weights with the eventual goal of reducing GFLOP computations. The pruning strategy is well motivated using the taylor expansion of the neural network function with respect to the feature activations. The obtained strategy removes feature maps that have both a small activation and a small gradient (eqn 7). 

(A) Ideally the gradient of the output with respect to the activation functions should be 0 at the optimal, but as a result of stochastic gradient evaluations this would practically never be zero. Small variance in the gradient across mini-batches indicates that irrespective of input data the specific network parameter is unlikely to change - intuitively these are parameters that are closer to convergence. Parameters/weights that are close to convergence and also result in a small activation are intuitively good candidates for pruning. This is essentially what eqn 7 conveys and is likely to be reason why just removing weights that result in small activations is not as good of a pruning strategy (as shown by results in the paper). There are two kind of differences in weights that are removed by activation v/s taylor expansion:
1. Weights with high-activations but very low gradients will be removed by taylor expansion, but not by activation alone. 
2. Weights with low-activation but high gradients will be removed by activation criterion, but not by taylor expansion. 
It will be interesting to analyze which of (1) or (2) contribute more to the differences in weights that are removed by the taylor expansion v/s activation criterion. Intuitively it seems that weight that satisfy (1) are important because they are converged and contribute significantly to network's activation. It is possible that a modified criterion - eqn (7) + \lambda feature activation, (where \lambda needs to be found by cross-validation) may lead to even better results at the cost of more parameter tuning. 
  
(B) Another interesting comparison is with the with the optimal damage framework - where the first order gradients are assumed to be zero and pruning is performed using the second-order information (also discussed by authors in the appendix). Critically, only the diagonal of the Hessian is computed. There is no comparison with optimal damage as authors claim it is memory and computation inefficient. Back of envelope calculations suggest that this would result only in 50% increase in memory and computation during pruning, but no loss in efficiency during testing. Therefore from a standpoint of deployment, I don't think this missing comparison is justified. 

(C) The eventual goal of the authors is to reduce GFLOPs. Some recent papers have proposed using lower precision computation for this. A comparison in GFLOPs with lower precision v/s pruning would be a great. While both these approaches are complementary and it is expected that combining both of them can lead to superior performance than either of the two - it is unclear when we are operating in the low-precision regime how much pruning can be performed. Any analysis on this tradeoff would be great (but not necessary).

(D) On finetuning, authors report results of AlexNet and VGG on two different datasets - Flowers and Birds respectively. Why is this the case? It would be great to see the results of both the networks on both the datasets. 

(E) Authors report there is only a small drop in performance after pruning. Suppose the network was originally trained with N iterations, and then M finetuning iterations were performed during pruning. This means that pruned networks were trained for N + M iterations. The correct comparison in accuracies would be if we the original network was also trained for N + M iterations. In figure 4, does the performance at 100% parameters reports accuracy after N+M iterations or after N iterations? 

Overall I think the paper is technically and empirically sound, it proposes a new strategy for pruning:
(1) Based on taylor expansion
(2) Feature normalization to reduce parameter tuning efforts. 
(3) Iterative finetuning. 
However, I would like to see some comparisons mentioned in my comments above. If those comparisons are made I would change my ratings to an accept.

[Author Response · Pavlo Molchanov · 14 Jan 2017 (modified: 16 Jan 2017)]
**New revision**

We would like to thank reviewers for their comments and suggestions. We added a new revision which addresses main points requested for clarification.
List of changes:
- Added comparison with Optimal Brain Damage. See section 2.2, paragraph "Relation to Optimal Brain Damage" for discussion, appendix A.6 for implementation details, Tables 1 and 3, Figures 4 and 5 for results and comparisons
- Comparison with average percentage of zeros criterion (APoZ) proposed by Hu et al. (2016)
- Comparison with pruning by regularization in appendix A.4
- Wall-clock time measurements for inference of pruned networks in section 3.6
- Results of combining Taylor and activation criteria in appendix A.5
- (Jan 16) Correlation of Taylor criterion with gradient and activation, appendix A.7

[Public Comment · (anonymous) · 02 Feb 2017]
**Some related works**

1) Wen, Wei, et al. "Learning structured sparsity in deep neural networks." Advances in Neural Information Processing Systems. 2016.
2) Lebedev, Vadim, and Victor Lempitsky. "Fast convnets using group-wise brain damage." Proceedings of the IEEE Conference on Computer Vision and Pattern Recognition. 2016.
3) Alvarez, Jose M., and Mathieu Salzmann. "Learning the Number of Neurons in Deep Networks." Advances in Neural Information Processing Systems. 2016.

[Final Decision · Program Chairs · 06 Feb 2017]
**ICLR committee final decision**

The paper presents a method for pruning filters from convolutional neural networks based on the first order Taylor expansion of the loss change. The method is novel and well justified with extensive empirical evaluation.